# An In Situ Study on Nanozyme Performance to Optimize Nanozyme-Strip for Aβ Detection

**DOI:** 10.3390/s23073414

**Published:** 2023-03-24

**Authors:** Yaying Luo, Haiming Luo, Sijia Zou, Jing Jiang, Demin Duan, Lei Chen, Lizeng Gao

**Affiliations:** 1School of Life Sciences, Division of Life Sciences and Medicine, University of Science and Technology of China, Hefei 230026, China; 2CAS Engineering Laboratory for Nanozyme, Institute of Biophysics, Chinese Academy of Sciences, Beijing 100101, China; 3MoE Key Laboratory for Biomedical Photonics, School of Engineering Sciences, Huazhong University of Science and Technology, Wuhan 430074, China; 4School of Basic Medical Sciences, Southwest Medical University, Luzhou 646000, China

**Keywords:** nanozyme-strip, peroxidase-like activity, NC membrane, in situ characterization, Aβ detection

## Abstract

The nanozyme-strip is a novel POCT technology which is different from the conventional colloidal gold strip. It primarily utilizes the catalytic activity of nanozyme to achieve a high-sensitivity detection of target by amplifying the detection signal. However, previous research has chiefly focused on optimizing nanozyme-strip from the perspective of increasing nanozyme activity, little is known about other physicochemical factors. In this work, three sizes of Fe_3_O_4_ nanozyme and three sizes of CoFe_2_O_4_ nanozyme were used to investigate the key factors of nanozyme-strip for optimizing and improving its detection performance. We found that three sizes of Fe_3_O_4_ nanozyme all gather at the bottom of the nitrocellulose (NC) membrane, and three sizes of CoFe_2_O_4_ nanozyme migrate smoothly on the NC membrane, respectively. After color development, the surface of NC membranes distributed with CoFe_2_O_4_ peroxidase nanozymes had significant color change. Experimental results show that CoFe_2_O_4_ nanozymes had better dispersity than Fe_3_O_4_ nanozymes in an aqueous solution. We observed that CoFe_2_O_4_ nanozymes with smaller particle size migrated to the middle of the NC membrane with a higher number of particles. According to the results above, 55 ± 6 nm CoFe_2_O_4_ nanozyme was selected to prepare the nanozyme probe and achieved a highly sensitive detection of Aβ42Os on the nanozyme-strip. These results suggest that nanozyme should be comprehensively evaluated in its dispersity, the migration on NC membrane, and the peroxidase-like activity to determine whether it can be applied to nanozyme-strip.

## 1. Introduction

Lateral flow immunoassay (LFIA) is a paper-based point-of-care testing (POCT) diagnostic tool, also known as test strips, which has been widely used for rapid detection and early screening of many diseases [1,2,3,4]. Compared with laboratory tests such as enzyme-linked immunosorbent assay (ELISA) and polymerase chain reaction (PCR), LFIA does not require automated instruments and professional testing personnel, and has the advantages of simple operation, rapid detection, and low cost. LFIA has been applied in many fields such as disease, agriculture, food, and so on. The most established LFIA is the colloidal gold nanoparticle-based paper strip (AuNP-strip) [5,6]. However, since the detection sensitivity of the AuNP-strip is significantly lower than those of laboratory assays, the reference value of its detection results is greatly reduced, which limits its application in disease monitoring. Upon on the above problems, many researchers have introduced nanozymes into paper strips in recent years [7,8,9,10,11] to replace the colloidal gold nanoparticles and developed nanozyme-strip with high sensitivity.

Nanozymes are a class of artificial nanomaterials with similar catalytic activity to natural enzymes [12,13,14,15], which are classified into four major categories based on their natural enzyme-like activities: oxidoreductases, hydrolases, polymerases, and isomerases [14]. Nanozymes possess superiorities due to their low cost, high stability, and easy preparation. Nowadays, nanozyme-strips mainly utilize the higher peroxidase (POD) activity of peroxidase nanozymes to generate colored products by catalyzing the oxidation of substrates, and accumulate in the detection zone to amplify the detection signal and enhance the detection sensitivity. Since the rapid detection of the Ebola virus using nanozyme-strip was first published by Yan teams in 2015 [16], many works on nanozyme-strip have been published [11,17,18]. The detection limit of the paper strip for procalcitonin reached 0.5 pg/mL, and its sensitivity was 2280-fold higher than that of the AuNP-strip [17].

Alzheimer’s disease is an irreversible neurodegenerative disease most common in the elderly. Amyloid β (Aβ) is considered to be the earliest detectable preclinical biomarker associated with the pathogenesis of AD [19]. Aβ consists of 38–43 amino acid residues derived from amyloid precursor protein that is sequentially cleaved by β- and γ-secretases, of which Aβ42 is the most neurotoxic [20]. Currently, abnormal Aβ42 levels can be detected in CSF and Aβ-PET neuroimaging [21,22]. However, these methods are difficult to be widely used because of high testing cost, invasiveness, and equipment requirements. It has been reported that the level of Aβ42 oligomers (Aβ42Os) reflects the level of Aβ42O in CSF and is highly correlated with the progression of AD, and thus has the potential to diagnose AD [23]. Based on the above, we are determined to take advantage of nanozyme-strips for rapid detection of Aβ42Os.

The nitrocellulose (NC) membrane is the carrier for the immunoreaction of nanozyme-strip [24]. After the detected target is specifically bound to the nanozyme probe as an antigen, it migrates forward on the NC membrane in the form of a complex. When it moves to the area where the capture antibody is immobilized, the complex is trapped by the antigen-antibody reaction. Here, the detection is finally realized by forming a test line (T line) and a quality control line (C line) through color development (Figure 1). As a result, the migration state of the nanozyme probe on the NC membrane is crucial to the final detection effect. However, the current design of nanozyme-strip focuses on how to improve the peroxidase-like activity of the nanozyme probe, while its flow state on the NC membrane has not been systematically studied. Factors such as size, morphology, and dispersion of nanozymes may affect their migration on the NC membrane and have a poor impact on the detection results.

In this article, CoFe_2_O_4_ and Fe_3_O_4_ nanozymes with different particle sizes synthesized in the laboratory were used to explore the factors affecting the migration of nanozyme probes on test strips, and the most suitable nanozyme was selected to prepare detection probes. By visualizing the distribution of nanozymes on NC membranes, we found that well-dispersed nanozymes do not aggregate during migration on NC membranes; the smaller the particle size, the better their uniform distribution on NC membranes. The POD activity of a nanozyme affects the color development of a nanozyme on the NC membrane. Finally, we comprehensively selected the 55 ± 6 nm CoFe_2_O_4_ peroxide nanozyme which had the best performance on the NC membrane for the preparation of nanozyme probes, and successfully detected Aβ42 oligomers (Aβ42Os), the blood marker of Alzheimer’s disease (AD) on the nanozyme-strip [25].

## 2. Materials and Methods

### 2.1. Reagents and Chemicals

Ferric chloride (FeCl_3_·6H_2_O), cobalt chloride hexahydrate (CoCl_2_·6H_2_O), ethylene glycol, diethylene glycol, 3,3′,5,5′-tetramethylbenzidine (TMB), sodium acetate trihydrate (NaAc·3H_2_O), and dimethyl sulfoxide (DMSO) were purchased from Sigma (Los Angeles, CA, USA). A nitrocellulose membrane was brought from Sartorius (Göttingen, Germany). Detachable 96-well plate, and bovine serum albumin (BSA) were purchased from Lablead Biotechnology Co., Ltd. (Beijing, China); 30% H_2_O_2_ was purchased from Sinopharm Chemical Reagent Co., Ltd. (Shanghai, China). The DAB color development kit was purchased from ZSGB-Bio (Beijing, China). Anti-beta Amyloid antibody was brought from Abcam (Cambridge, UK). The 1F12 monoclonal antibody was supplied by courtesy of Prof Haiming Luo’s Lab in the Huazhong University of Science and Technology (Wuhan, China).

### 2.2. Instrumentation

An ultraviolet (UV) spectrophotometer (Beckham Coulter, Brea, CA, USA) was utilized to measure the absorption of TMB working solution with nanozymes at 652 nm. Particle size and Zeta Potential Analyzer (Brookhaven, New York, NY, USA) was applied to measure the value of Zeta potential and PDI. X-ray powder diffractometer (Bruker, Billerica, MA, USA) was used to identify the composition of the synthesized nanozymes. An infrared spectrometer (Thermo Scientific Nicolet, Waltham, MA, USA) was applied to characterize the modified groups on the surface of the nanozymes. The Field Emission Transmission Electron Microscopy (FEI Company, Hillsboro, OR, USA) was utilized to analyze the morphology and elemental mapping of the nanozymes. The Field Emission Scanning Electron Microscopy (Hitachi, Tokyo, Japan) was used to observe the distribution of nanozyme particles on the NC membrane.

### 2.3. Preparation and Characterization of Nanozymes

CoFe_2_O_4_ peroxidase nanozymes with three particle sizes were synthesized by adjusting the input amount of CoCl_2_·6H_2_O following a modified solvothermal method. Briefly, 3.75 g maleic acid was fully dissolved in 40 mL of ethylene glycol. After stirring overnight, 1.35 g FeCl_3_·6H_2_O, a certain amount of CoCl_2_·6H_2_O, and 3.75 g NaAc were successively added into the mixed solution. After being stirred vigorously and ultrasonically for 1 h, the mixture was transferred into a Teflon-lined stainless-steel autoclave and reacted at 200 °C for 12 h. Then, the obtained products were washed alternately by ethanol and deionized water for six times, and finally reserved in water. To synthesis different sizes of CoFe_2_O_4_, the amount of added CoCl_2_·6H_2_O were 0.45 g, 0.6 g, and 0.9 g.

Fe_3_O_4_ peroxidase nanozymes with three particle sizes were synthesized by adjusting the input amount of FeCl_3_·6H_2_O following a modified solvothermal method. Briefly, a certain amount of FeCl_3_·6H_2_O was dissolved in 40 mL ethylene glycol. Next, 3.6 g NaAc was added into the mixed solution. After be stirring vigorously and ultrasonically for 0.5 h, the mixture was transferred into a Teflon-lined stainless-steel autoclave and reacted at 200 °C for 12 h. Finally, the obtained products were washed alternately by ethanol and deionized water for six times, and dried under a −50 °C vacuum for 12 h. To synthesis different sizes of Fe_3_O_4_, the amount of added FeCl_3_·6H_2_O were 0.6 g, 0.8 g, and 1.8 g.

Transmission electron microscopy (TEM), scanning electron microscopy (SEM), X-ray crystal diffraction (XRD), energy-dispersive X-ray spectroscopy (EDS), X-ray photoelectron spectroscopy (XPS), and Fourier transform infrared spectroscopy (FTIR) were used to characterize the synthesized nanozymes.

### 2.4. Exhibition of Migration State of Nanozymes on NC Membrane

Referring to the specific parameters of the NC membrane: the width is 2.5 cm, the chromatography speed is 134.3 s/4 cm, and the chromatography time of the NC membrane insertion hole is set to 84 s. Prepare different concentrations of nanozyme solutions with deionized water: 0.1 mg/mL, 0.2 mg/mL, 1 mg/mL; fully sonicate the above nanozyme solution and shake before use to ensure that there is no precipitation in the solution. A total of 70 μL of nanozyme solution was pipetted, then added to the well plate and the NC membrane was immediately inserted. After 84 s, the NC membrane was taken out and completely immersed in the chromogenic solution (DAB solution or TMB solution). After 5 min, the developed NC membrane was taken out and the surface color change was recorded by taking pictures with a smart phone. The NC membrane we used was attached to a polyvinyl chloride (PVC) plate and evenly cut into 3 mm width strips. All the steps were conducted at room temperature.

### 2.5. Determination of Specific Activity of Nanozymes

To prepare different concentrations of nanozyme aqueous solution, the reaction system consists of 1.8 mL pH 4.5 0.2 mol/L NaAc buffer solution, 0.1 mL nanozyme solution of a certain concentration, 0.1 mL 10 mg/mL TMB solution (DMSO dissolved), and 0.2 mL 10 mol/L H_2_O_2_. The above solutions were added sequentially in a quartz cuvette, and the change in absorbance of the reaction solution at 652 nm within 1 min was measured using a UV spectrophotometer, and the specific activity of the nanozyme was calculated by graphing [26]. All the measurements were conducted at 37 °C.

### 2.6. Determination of Zeta Potential and PDI of Nanozymes

To measure the zeta potential, 0.01 mg/mL nanozyme solution was prepared with deionized water, and 1.6 mL of the solution was pipetted into a cuvette after 5 min of ultrasonication, followed by insertion of the electrode, and the cuvette was placed in the particle size and zeta potential analyzer. Each sample was measured for 3 cycles. PDI was measured similarly, and 0.01 mg/mL nanozyme solution was prepared with deionized water. After 5–10 min of ultrasonication, 1.8 mL of the solution was immediately pipetted into a cuvette, and subsequently placed in the particle size and zeta potential analyzer. Each sample was measured for 3 cycles. All the above measurements were conducted at 25 °C.

### 2.7. Observation of Nanozyme Distributed on NC Membrane by SEM

A total of 70 μL of 0.1 mg/mL nanozyme solution was added into the well plate and the NC membrane was quickly inserted. After 84 s, the NC membrane was taken out and transferred to a 37 °C oven for drying. After drying completely, SEM was used to observe the nanozyme particles at the bottom and middle of the NC membrane, and the energy disperse spectroscopy (EDS) element scanning was used to scan the middle of the NC film for Fe and Co elements.

### 2.8. Preparation of 1F12-Conjugated Carboxyl-Modified Nanozymes

The nanozyme probes were synthesized by the formation of an amido bond [27,28] between the residual carboxyl group of the nanozyme and the amino group of 1F12 mAb. Briefly, 500 μg peroxide nanozyme were first activated with 5 mM EDC and NHS solution (50 mM MES, pH 5.0). After incubation for 30 min at room temperature, the supernatant was removed by magnetic separation. The nanozyme was resuspended by adding 500 μL 0.1 mg/mL 1F12 antibody solution, mixed thoroughly, and reacted overnight at 4 °C. The mixture was removed from the supernatant by magnetic separation and 500 μL, 50 mM pH 7.4 Tris-HCl was immediately added. After shaking at room temperature for 30 min, 5% BSA (10 mM PBS) was added to block the non-specific binding sites on the nanozyme surface. After incubation for 60 min at room temperature, the supernatant was removed by magnetic separation and 500 μL of PBS (10 mM pH 7.4, 0.01% Tween20) was added to resuspend the precipitate, at which time the concentration of the resulting probe was 1 mg/mL, The probe solution was stored at 4 °C for further use.

### 2.9. Assembly of Nanozyme-Strip

The main components of the nanozyme-strip used include NC membrane, absorbent pad and polyvinyl chloride (PVC) plate. The recombinant Anti-beta Amyloid antibody and goat anti-mouse IgG were immobilized on the NC membrane to form the detection area and quality control area, and dried at 37 °C for 2 h. Firstly, the antibody-coated NC membrane was attached to a PVC plate, and the absorbent pad was then placed on the top of the NC membrane by overlapping 2 mm. After assembly, the plate was evenly cut into 3 mm width strips, sealed with a desiccant, and stored at room temperature.

### 2.10. Detection of Aβ42Os on Nanozyme-Strip

A range of standard solutions of Aβ42Os was prepared by diluting the Aβ42Os stock solution with 50 mM TBS. When testing, 70 μL of antigen solution and 2 μL of nanozyme probe was mixed in the sample well and the test strip was inserted. After 10 min of chromatography, the reacted test strip was taken out and immersed in a commercial DAB chromogenic solution. After immersion for 5 min, the reacted test strip was washed with water for several times, and subjected to qualitative and quantitative measurements. The photographs of the test strips were recorded by a smartphone. We converted the color of T lines into the grey value by ImageJ, then constructed the linear relation between the concentration of Aβ42Os and the signal intensity.

## 3. Results

### 3.1. Characterization of Fe_3_O_4_ and CoFe_2_O_4_ Nanozymes

Fe_3_O_4_ nanozyme [12] and CoFe_2_O_4_ nanozyme [11] with different sizes were synthesized by the solvothermal method in accordance with previous reports with slight modifications, and their morphologies were observed by TEM and SEM. TEM results show that three particle sizes of Fe_3_O_4_ nanozymes were synthesized: 34 ± 6 nm Fe_3_O_4_-1 (*n* = 30) (Figure 1a), 64 ± 9 nm Fe_3_O_4_-2 (Figure 1b), and 184 ± 12 nm Fe_3_O_4_-3 (Figure 1c); and three particle sizes of CoFe_2_O_4_ nanozymes: 55 ± 6 nm CoFe_2_O_4_-1 (Figure 1d), 91 ± 7 nm CoFe_2_O_4_-2 (Figure 1e), and 146 ± 10 nm CoFe_2_O_4_-3 (Figure 1f). The morphology of Fe_3_O_4_-1 (Figure 1g) and Fe_3_O_4_-2 (Figure 1h) were irregularly spherical and heavily aggregated, and Fe_3_O_4_-3 (Figure 1i) was spherical; SEM show that CoFe_2_O_4_-1 (Figure 1j), CoFe_2_O_4_-2 (Figure 1k), and CoFe_2_O_4_-3 (Figure 1l) were spherical. The particle size distributions were shown in Appendix A.

Synthesized Fe_3_O_4_ nanoparticles and CoFe_2_O_4_nanoparticles were both analyzed by X-ray crystal diffraction (XRD). The XRD spectra of different particle sizes of Fe_3_O_4_ in Figure 2a matched the Fe_3_O_4_ standard card exactly. Moreover, the energy-dispersive X-ray spectroscopy (EDS) mapping (Figure 2j–m) and X-ray photoelectron spectroscopy (XPS) were also measured to verify the successful synthesis of Fe_3_O_4_ (Appendix A), and the results indicate that the synthesized nanoparticles were all Fe_3_O_4_. Figure 2b shows that the XRD patterns of the synthesized CoFe_2_O_4_ can well match with the CoFe_2_O_4_ standard card, except that the characteristic peak at about 55° is not distinct. Local analysis of the XRD patterns of CoFe_2_O_4_ shows that there was a weak diffraction peak at approximately 55°. Compared with the diffraction peaks of Fe_3_O_4_, the diffraction peaks of CoFe_2_O_4_ are significantly broadened, attributing to the difference in the ionic radius of Co and Fe ions. Moreover, the diffraction peak at about 43° is not shown in the XRD patterns of CoFe_2_O_4_ due to the broadening effect of the diffraction peak at 41°. These results indicates that Co is involved in the synthesis of CoFe_2_O_4_. The energy-dispersive X-ray spectroscopy (EDS) mapping (Figure 2e–j) demonstrates the presence of Co in CoFe_2_O_4_ nanozyme, and X-ray photoelectron spectroscopy (XPS) was further used to prove the successful synthesis of CoFe_2_O_4_ (Appendix A).

NaAc was added during the synthesis process in order to make the nanoparticles with carboxyl groups, Fourier transform infrared spectrum (FTIR) was utilized to analyze the modified groups on the surface of CoFe_2_O_4_ and Fe_3_O_4_ nanozymes. In Figure 2c, the vibrational peaks near 1560 cm^−1^ and 3410 cm^−1^ for CoFe_2_O_4_ nanozyme correspond to C=O and -OH vibrations, respectively, indicating the presence of carboxyl groups on the surface of CoFe_2_O_4_ nanoparticles; Figure 2d shows that the vibrational peaks of Fe_3_O_4_ nanozyme near 1630 cm^−1^ and 3400 cm^−1^ correspond to C=O and -OH vibrations, respectively, indicating the presence of carboxyl group modification on the surface of Fe_3_O_4_ nanoparticles as well [29,30].

### 3.2. Characterization of Nanozymes on NC Membrane by Color Development

The NC membranes distributed with CoFe_2_O_4_ nanozymes were developed in DAB and TMB solution, and had significant color change on their surface. With increasing nanoparticle concentration: 0.1 mg/mL, 0.2 mg/mL, and 1 mg/mL (Figure 3b,c), the color of NC membranes had a tendency to deepen. In Fe_3_O_4_ group, a large number of nanoparticles can be observed on the bottom part of the NC membranes before color development, and there was almost no color change on its surface after color development (Figure 3b,c) compared with control NC membrane, indicating that CoFe_2_O_4_ nanozymes performed better on NC membrane. When the concentration of CoFe_2_O_4_ solution increased to 1 mg/mL, the color of the nanoparticles largely overwhelmed the color of the blue oxidation products of TMB, the phenomenon might affect the signal amplification of detection on nanozyme-strip. Moreover, the color development effect of DAB on the NC membrane became more and more distinct with the increasing concentration of the nanozyme solution, indicating that a better effect on NC membrane was achieved by using DAB. Hence, we chose DAB as the chromogenic solution in the later nanozyme-strip assay.

The POD activities of the two nanozymes were compared, and Figure 4a shows that the OD652 nm values of TMB reaction catalyzed by different particle sizes of CoFe_2_O_4_ peroxide nanozymes were in the range of 2.53–2.60 at 285 s, while the OD652 nm values of different particle sizes of Fe_3_O_4_ peroxide nanozymes were in the range of 0.16–0.25. Further, their POD-specific activities were determined [26], the POD-specific activity of CoFe_2_O_4_-1, CoFe_2_O_4_-2, and CoFe_2_O_4_-3 were 5.929 U/mg, 5.511 U/mg, and 5.165 U/mg, respectively; and the POD-specific activity of Fe_3_O_4_-1, Fe_3_O_4_-2, and Fe_3_O_4_-3 were 0.404 U/mg, 0.483 U/mg, and 0.292 U/mg, respectively. The results show that the POD-specific activity of CoFe_2_O_4_ was significantly higher than that of Fe_3_O_4_ (Figure 4b).

Zeta potential is an important indicator to characterize the stability of a dispersion system. We measured the zeta potentials of CoFe_2_O_4_ and Fe_3_O_4_ nanozymes separately. The zeta potentials of CoFe_2_O_4_-1, CoFe_2_O_4_-2, and CoFe_2_O_4_-3 were −40.93 mV, −35.75 mV, and −40.55 mV, respectively; the zeta potentials of Fe_3_O_4_-1, Fe_3_O_4_-2, and Fe_3_O_4_-3 were −14.68 mV, −13.79 mV, and −19.82 mV, respectively. Figure 2e revealed that the absolute values of zeta potential of CoFe_2_O_4_ group were significantly higher than those of Fe_3_O_4_ group. These results indicate that CoFe_2_O_4_ nanozymes had better dispersion than Fe_3_O_4_ nanozymes. Meanwhile, their polymer dispersion index (PDI) was measured, and the PDI of CoFe_2_O_4_ nanoparticles is in the range of 0.05~0.06, while that of Fe_3_O_4_ nanoparticles is in the range of 0.25~0.27 (Appendix A).

### 3.3. Characterization of CoFe_2_O_4_ Nanozymes on NC Membrane by SEM

The surface of the undeveloped NC membranes distributed with CoFe_2_O_4_ nanoparticles were observed by SEM, and each of its bottom and middle parts were observed. Compared with the blank NC membrane (Figure 5a,b), Figure 5h shows that the bottom of CoFe_2_O_4_-3-NC membrane has a large number of nanoparticles (pointed out by the red arrow) distributed, while the middle part (Figure 5g) has very few nanoparticles distributed compared with the CoFe_2_O_4_-3-NC membrane. Both CoFe_2_O_4_-1-NC membrane and CoFe_2_O_4_-2-NC membrane have fewer nanoparticles at the bottom (Figure 5d,f), but more nanoparticles at the middle part of membranes (Figure 5c,e). The surface of the undeveloped NC membranes distributed with Fe_3_O_4_ nanoparticles were also observed by SEM. Compared with the blank NC membrane, numerous Fe_3_O_4_ nanoparticles were aggregated at the bottom part of the NC membranes (Appendix A), and at the middle part of the NC membranes it was hard to find nanoparticles (Appendix A). The results indicate that the NC membrane had color change in the distance from the bottom to the black belt when the concentration of Fe_3_O_4_ solution was 1 mg/mL. These results indicate that Fe_3_O_4_ nanoparticles failed to migration forward after the aggregation occurred on the NC membrane.

EDS elemental surface scan was performed on the middle part of the undeveloped NC membranes for Fe and Co elements (Figure 6a), and the average signal intensity of each Merge image was analyzed using ImageJ. Figure 6b shows that the average signal intensities of the nanozymes distributed on the surface of CoFe_2_O_4_-1-NC, CoFe_2_O_4_-2-NC, and CoFe_2_O_4_-3-NC membranes were 50.413, 50.267, and 34.104, respectively. The average signal intensities of Fe and Co elements in the middle of the CoFe_2_O_4_-3-NC membrane were significantly lower than those of CoFe_2_O_4_-1-NC and CoFe_2_O_4_-2-NC membranes, suggesting that the size of nanoparticles may affect their migration on the NC membrane. This result was consistent with the SEM observation. Combined with the results of nanoparticle distribution on NC membrane characterized by DAB solution (Figure 3b), CoFe_2_O_4_-1 has better color development effect than CoFe_2_O_4_-2 on NC membrane and was eventually chosen for the preparation of nanozyme probes.

### 3.4. Detection of Aβ42Os with Nanozyme Test Probe

The nanozyme probe for detecting Aβ42 oligomers (Aβ42Os) was prepared by using the CoFe_2_O_4_-1 nanozyme that performed best on the NC membrane, and its performance on the nanozyme-strip was tested. Aβ42Os is a biomarker of AD, and Aβ42Os in blood has the potential to diagnose AD. The oligomer form in the Aβ42Os stock solution used in the experiment is mainly a trimer. Figure 7a depicts the principle of CoFe_2_O_4_-1 nanozyme probe for highly sensitive detection of Aβ42Os. Figure 7b shows that the nanozyme probe can detect Aβ42Os in the concentration range of 0.18–1820 ng/mL, and there was no retention of the nanozyme probe at the bottom of the NC membrane. In order to evaluate the performance of Aβ42Os testing by nanozyme-strip, the signal intensity of T line was analyzed by ImageJ. There was a good correlation (R^2^ = 0.9796) between the T line signal intensity and the detection concentration within the concentration range of 0.18–1820 ng/mL (Figure 7c). Previous studies have reported that the Aβ42Os levels in the blood of AD patients range from 0.1 to 1 ng/mL, as detected by sandwich ELISA [31]. The sensitivity of the nanozyme-strip for Aβ42Os detection needs to be improved.

## 4. Discussion

The nanozyme applied to the nanozyme-strip needs to migrate smoothly on the NC membrane without aggregation during the lateral movement, so that more nanozyme probes can reach the detection area and achieve the high-sensitivity detection of target [32].

To investigate the migration of CoFe_2_O_4_ and Fe_3_O_4_ nanozymes on NC membrane, the NC membranes distributed with the nanozymes were developed by peroxidase substrate solution (DAB or TMB). The nanozymes with POD-like enzyme activity can catalyze the peroxidase substrates to colored products which deposit on the surface of the NC membranes, leading to a significant color change (the oxidation of DAB produces a brown color product; the oxidation of TMB produces a blue color product). The color of NC membranes in CoFe_2_O_4_ group was much deeper than that in Fe_3_O_4_ group, and the NC membrane distributed with CoFe_2_O_4_-1 nanozymes had the darkest color under the same concentration (Figure 3b). The POD activity of CoFe_2_O_4_ nanozymes with different particle sizes had no significant difference (Figure 4c). A possible explanation might be that the largest number and even distribution of CoFe_2_O_4_-1 nanoparticles per unit area makes the NC membrane the deepest after color development. In previous studies of the nanozyme-strip, researchers mainly focused on the activity of nanozymes. Our study shows that enzyme-like activity should not be the only factor to be considered but also the dispersibility of the nanozyme should be taken into account. Zeta potential is an important parameter to reflect the charged state of the particle surface in the suspension, which can be used to analyze the stability of nanoparticles in the dispersion system [33]. The absolute value of the Zeta potential of CoFe_2_O_4_ nanoparticles with three particle sizes was significantly higher than that of Fe_3_O_4_ nanoparticles with three particle sizes. However, there is no significant difference in the absolute value of Zeta potential within the group. Owing to the low absolute value of the zeta potential, the attraction between Fe_3_O_4_ particles is greater than the repulsive force, resulting in the aggregation of nanoparticles and failing to move on the NC membrane after migrating a short distance.

In order to compare the distribution of CoFe_2_O_4_ nanozymes with three particle sizes on the NC membrane, the surface of the NC membranes was observed by SEM. SEM shows that there were a large number of nanoparticles distributed at the bottom of the CoFe_2_O_4_-3-NC membrane and very few in the middle zone, which demonstrated that the CoFe_2_O_4_-3 distribution on the CoFe_2_O_4_-3-NC membrane was uneven and mostly distributed at the bottom. Further, EDS was used to scan the middle part of these NC membranes for Fe and Co elements, the signal intensity of Fe and Co elements on CoFe_2_O_4_-3-NC membrane was significantly lower than that of CoFe_2_O_4_-1-NC membrane and CoFe_2_O_4_-2-NC membrane, indicating that a fewer number of CoFe_2_O_4_-3 nanoparticles moved to the middle of the NC membrane. We speculated that the migration of nanozymes with good dispersity would be also affected when the nanoparticle size reaches a certain level. The distribution of CoFe_2_O_4_-1 and CoFe_2_O_4_-2 on NC membrane did not show significant differences, but the NC membrane distributed with CoFe_2_O_4_-1 had the darkest color. We selected CoFe_2_O_4_-1 nanozyme, which performed the best on the NC membrane from the candidates to prepare the nanozyme probe for detecting Aβ42Os by a comprehensive evaluation of three aspects: dispersibility, distribution on the NC membrane, and the level of POD activity. In the presence of Aβ42Os, the CoFe_2_O_4_-1 nanozyme probe did not show aggregation and retention on the NC membrane, T lines and C lines appeared on its surface after color development. The experimental results showed that the probe can migrate forward smoothly on the NC membrane (Figure 7b), and the detection of Aβ42Os was realized. At the same time, for the detection of Aβ42Os, more optimization studies are needed to improve the sensitivity of the nanozyme-strip. In summary, the selected CoFe_2_O_4_-1 nanozyme can be applied to the nanozyme-strip, and has the potential to be widely used to prepare nanozyme probes for different targets.

The Fe_3_O_4_ nanozyme with poor dispersion only migrated a short distance on the NC membrane before aggregation occurred, and failed to move to the detection area on the NC membrane; whereas CoFe_2_O_4_ nanozymes had better dispersion and could migrate forward on the NC membrane, and a large amount of CoFe_2_O_4_ nanozymes could reach the detection area of the NC membrane. Although the particle size of Fe_3_O_4_-1 nanoparticles is smaller than that of CoFe_2_O_4_-1 nanoparticles, poor dispersion of Fe_3_O_4_-1 nanoparticles led to its aggregation during migration on the NC membrane, and it could not migrate forward as smoothly as CoFe_2_O_4_-1 nanoparticles, resulting in a vast number of nanoparticles accumulated at the bottom of the NC membrane. Hence, we considered that dispersity is the most important parameter for nanozyme applied to nanozyme-strip. Dispersity affects the movement of nanozymes on the NC membrane. The nanozyme probe prepared by nanozymes with better dispersion migrate more to the detection zone, which is beneficial for the results. For nanozymes with good dispersion, we can optimize them in terms of particle size and enzyme-like activity to further improve the performance of the nanozyme-strip.

Meanwhile, both CoFe_2_O_4_ nanozyme and Fe_3_O_4_ nanozyme have been reported to show superparamagnetic properties [11,12]. Nanozymes with magnetic response can be used to enrich the analytes in the sample to improve the detection sensitivity of the nanozyme-strip, and the magnetic separation of nanozymes can also simplify the preparation of nanozyme probes. Nanozymes with magnetic properties and good dispersity are ideal for nanozyme probes. The study of nanozyme-strips should focus on the synthesis of nanozymes with such properties.

## 5. Conclusions

We optimized the nanozyme-strip by using candidate nanozymes synthesized in the laboratory, focusing on the nanozyme migration state on the NC membrane. The migration state of the nanozyme probe on the NC membrane greatly affects the detection ability of the nanozyme-strip. Dispersity of the nanozyme dominates its migration state on the NC membrane. Zeta potential and PDI value of nanozymes can indirectly forecast its movement on the NC membrane: the higher the absolute value of the zeta potential (>35 mV), the smaller the PDI value (0.05 level), and the better dispersion of the nanozyme, the less likely it is to aggregate on the NC membrane. More attention should be paid to the dispersion of nanozymes when screening nanozymes to prepare the detection probe for the nanozyme-strip. The migration and distribution of nanozymes on the NC membrane and the enzyme-like activity should be comprehensively investigated, and the best-performed nanozyme is screened out among the candidate nanozymes for the preparation of detection probes so as to achieve high-sensitivity detection of the target, which can also save the use of antibodies and reduce the development cost of nanozymes-strips. The work of this paper aims to promote the research of nanozyme-strips.

## Data Availability

The data presented in this study are available on request from the corresponding author.

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
