# Peer review of "An In Situ Study on Nanozyme Performance to Optimize Nanozyme-Strip for Aβ Detection"

_sensors, 2023, doi:10.3390/s23073414_

Round 1

Reviewer 2 Report

In this work, the authors studied three sizes of Fe3O4 nanozyme and three sizes of CoFe2O4 nanozyme to investigate the key factors of nanozyme-strip for optimizing and improving its detection performance and gave an acceptable conclusion finally. I suggest its publication after addressing the following issues.

1.       In the abstract, the full name of NC membrane was missed.

2.       In Figure 2a and b, the name of nanozyme should be properly written.

3.       For the size of the nanozymes, particle size distribution should be given.

4.       Please explain the color change in Figure 3. Why it is mixed gray and bule in 1 mg/mL

Reviewer 3 Report

In this paper, the authors investigated the relationship between the physicochemical properties of nanozymes and nanozyme-strip performance. Nanozyme-strip is a promising technique for rapid and sensitive detection. Besides the catalytic activity, the authors investigated the nanozymes with different sizes, surfaces, and charge in nanozyme-strip using in situ characterization. Based on these characterizations, the authors screened out a CoFe2O4 nanozyme suitable for nanozyme-strip and used it to detect Aβ oligomer with high sensitivity. The work is important for developing reliable nanozyme-strip. Thus the manuscript merits the consideration of acceptance for publication on Sensors.

1.      In this paper, several parameters of nanozymes are investigated, which is the most important for nanozyme-strip? More discussion is needed.

2.      Both CoFe2O4 nanozyme and Fe3O4 nanozyme have superparamagnetic properties. Would such magnetic property help improve nanozyme-strip performance? The authors can give some discussion and prospects.

3.      The detection limit for Aβ oligomer is about 0.18 ng/ml. Can such sensitivity match the clinical test? What is the level of Aβ oligomer in the patient?

Round 2

Reviewer 1 Report

The authors have addresed all questions properly. I recommend this paper for its publication in Sensors.